# Differences in Coordination Motor Abilities between Orienteers and Athletics Runners

**DOI:** 10.3390/ijerph20032643

**Published:** 2023-02-01

**Authors:** Weronika Machowska-Krupa, Piotr Cych

**Affiliations:** Faculty of Physical Education and Sport, Wroclaw University of Health and Sport Sciences, 51-612 Wroclaw, Poland

**Keywords:** coordination motor abilities, orienteering, foot orienteering, athletics

## Abstract

This study aimed to examine the differences in coordination motor abilities between track and field (T&F) runners and foot orienteers (Foot-O). Another purpose of this study was to analyse gender differences in terms of coordination motor abilities. Coordination skills tests were undertaken by 11 Foot-O and 11 T&F runners. Each group consisted of five women and six men who lived in the Lower Silesia region of Poland. The Foot-O group consisted of 11 orienteers aged 24.09 (±4.78) years, with a minimum 10 years of experience, while the T&F group consisted of 11 long-distance runners aged 24.91 (±4.04) years and with a performance level at distances of 5 km and 10 km equivalent to that for orienteering. Some of the participants represented world-class level (e.g., world junior medallists), and most of them were of national elite level. Coordination tests of motor abilities were chosen for their reliability and repeatability and included tests of spatial orientation, rhythmisation of movements, balance and kinaesthetic differentiation. The Foot-O group performed significantly better than the T&F group in terms of some coordination abilities. Differences were observed between the Foot-O and T&F runners in balance ability measured during the “Walk on the bench” test. Further research should be carried out in this area in order to confirm these differences.

## 1. Introduction

Over the years, a wide variety of terms relating to motor abilities have been used by various disciplines. For instance, terminology such as motor proficiency, motor coordination, motor skills, motor ability, movement problems and movement skills have been used [1]. While important differences exist between these terms and conceptualisations, they are not always clearly defined in research. Consequently, the variety in nomenclature can be somewhat misleading. In this article, motor abilities are based on the definition created by Burton et al. [2]. In this context, motor abilities refer to a person’s potential movement competency and are described as not being directly observable, but are derived from a set of movement tasks. Motor abilities measure the general traits or capacities that underlie the performance of a wide variety of motor skills. For example, balance can be derived from a set of tasks, such as walking on a line, standing on a balance beam or standing on one leg [2].

Coordination is not only divided into spatial and temporal orders. Rather, it refers to different kinds and degrees of functional order among interacting parts and processes in both space and time. However, it is only in the last 25 years or so, and under quite peculiar circumstances, that basic laws for the quantitative description of coordination have been found [3].

Coordination of movements is of paramount importance for the effectiveness of any sports activity undertaken. That is why so much attention is being paid to its study, as well as time spent improving it. Each sport requires a different—peculiar—motor coordination, although there are also elements common to many sports such as balance, spatial orientation, ability to differentiate movements, etc. The influence of motor coordination on performance in sports has already been studied by many researchers in many fields [4,5,6]. Running is one of the most popular forms of sports activity. In running, it is the coordination and phase relationships between the action of the lower leg and the thigh in the sagittal plane that cause flexion and extension in the knee joint. In the frontal plane, the actions of the legs and feet must be coordinated to produce pronation and supination at the subtalar joint [7]. To understand the role of coordination in running, we need to look at the functional patterns of coordination within the lower limb as well as between all parts of the human body. The coordinated actions of lower extremity segments are also necessary to absorb the impact forces generated during running [7]. Moreover, Besson et al. [8] pointed to a number of existing differences in the running techniques of men and women. A number of sex differences have been identified in running related to biomechanical, physiological and neuromuscular aspects, some of which potentially play a role in endurance running performance. In particular, females show greater hip and knee joint motion in non-sagittal planes and greater lower limb muscle activation, compared to their male counterparts at a given absolute speed [8].

Orienteering is a sport that “involves an individual or team-based race in which the orienteer or teams of orienteers must, as rapidly as possible, find a series of control points in unfamiliar terrain” [9] (p. 6), using a map and compass [10,11]. While covering the route, athletes usually perform long-time activity lasting from 30 min to several hours [12], although the so-called sprint distances are an exception—the time of effort oscillates around several minutes. Orienteering is an endurance sport [13,14,15], which is highly beneficial for the development of the physical, cognitional and mental abilities of those who practice it [16,17,18]. Therefore, success in this discipline depends not only on physical fitness but also on intellectual prowess because the runner must constantly decide which way and how fast he will run, memorize the content of the map to be able to smoothly recall it from memory, quickly notice important elements in the field and on the map and make necessary adjustments in the implementation and planning of his action [19,20]. Competitors compete on a route usually leading through unpopulated terrain [21]. The terrain is often mountainous and forested, and usually with uneven pavements. It requires from runners the ability to maintain balance and agility so they can quickly change their direction of movement and avoid obstacles encountered [22]. Furthermore, “in orienteers, sports performance depends on a set of the integral and closely connected elements of cardio- and hemodynamic psychological status, coordination abilities and the speed of sensorimotor reactions” [23] (p. 1688). Movement coordination is very important in orienteering because the participant of training or competition has the need to constantly adjust his movements to changing conditions—ground, slope inclination, vegetation, etc. People practicing orienteering must learn fundamental movement skills such as running, jumping and climbing set in a forest environment. With these basic movements, children and adults will obtain confidence with agility, balance and coordination [24].

This study aimed to examine differences in coordination motor abilities between track and field (T&F) runners and foot orienteers (Foot-O). Differences in coordination between men and women practicing these two different sports were also analysed. The results of the study could indicate the need to train specific coordination skills for achieving sports mastery in a particular sport.

## 2. Materials and Methods

### 2.1. Participants

Diagnostic studies examined 22 participants who were stratified into two groups of 11. Each group consisted of five women and six men who lived in the Lower Silesia region of Poland. The Foot-O group consisted of 11 orienteers aged 24.09 (±4.78) years, with a minimum 10 years of experience, while the T&F group consisted of 11 long-distance runners aged 24.91 (±4.04) years and with a performance level at distances of 5 km and 10 km equivalent to that for orienteering. Some of the participants represented world-class level (e.g., world junior medallists) and most of them were of national elite level.

### 2.2. Measures

Examinations were conducted at the facilities of the University of Health and Sport Sciences in Wroclaw (Poland). The research consisted of an interview and physical tests of selected coordination skills. This study was approved by the Senate Committee on Ethics of Scientific Research of the University of Health and Sport Sciences in Wroclaw (Ref: 6/2013). The interview concerned gender, age and training internship details. Coordination tests of motor abilities were chosen for their reliability and repeatability according to Raczek et al. [25], and included temporal and spatial orientation, rhythmisation of movements, balance and kinaesthetic differentiation. The set of motor coordination tests was carried out in the following order:“Walk to the goal”;“Run at a given pace”;“Walk on the balance beam”;“Standing long jump from a place at 50% of maximum effort”.

### 2.3. Procedures

The first test was performed immediately after the subject’s arrival and explanation of the task. The subjects rested for 3 min between successive repetitions of the trials. The second test was preceded by an athletic warm-up (15 min) and a single attempt to run on hoops and over a distance of 30 m. The subjects accelerated to run from a gravel path 10 m away from the playing field to perform the maximum speed test. The subjects rested for 5 min between trials. All subjects were provided with appropriate conditions to test coordination of motor skills, as the research was carried out in good and windless weather.

After two outdoor tests, two more tests were carried out indoors after 10 min break. For these tests, the interval between repetitions was 5 min.

#### 2.3.1. “Walk to the Goal” Test

Spatial orientation ability was assessed using the “Walk to the goal” test [25] (pp. 156–157), during which the subject’s task was to reach the centre of a circle (1 m in diameter) that was placed 5 m away (Figure 1). Distance to the circle was measured in centimetres from the point between the feet of the subject and the centre of the circle. Additionally, subjects wore a black scarf around their eyes to exclude the sense of sight throughout the task. Each participant completed the test five times, and the result of the trial was calculated as “the average result of five trials in centimetres” [25] (p. 157). To make the measurement more accurate, according to the results of Thomson’s research [26], the repetitions were carried out in such a way that the subjects moved immediately after seeing the target and covering their eyes (up to 8 s).

#### 2.3.2. “Run at a Given Pace” Test

The rhythmisation of movements was assessed by the “Run at a given pace” test [25] (pp. 165–167). The test consisted of running a distance of 30 m in the shortest possible time. Then, a distance of 30 m had to be covered while inserting the feet into 11 hoops (60 cm in diameter), as shown in Figure 2. The difference in time between both trials was then calculated to determine the results. Each test began with an athletics warm-up, which was followed by a single test of both variations of the task for each participant. In order to perform the test at maximum speed, each participant used a 10 m stretch of a gravel path to accelerate to a run immediately before commencing the task.

#### 2.3.3. “Walk on the Balance Beam” Test

Balance ability was assessed by the “Walk on the balance beam” test [25] (p. 164). Subjects walked back and forth over a 10 cm wide and two-meter-long wooden balance beam (Figure 3) for 45 s. Each participant completed the task three times. Results were calculated as the average distance, expressed in meters, from the best two trials lasting 45 s or until the loss of balance.

#### 2.3.4. “Standing Long Jump from a Place at 50% of Maximum Effort” Test

“Standing long jump from a place at 50% of maximum effort” test [25] (pp. 152–153) was used to measure kinaesthetic differentiation ability. Each participant performed a standing long jump (Figure 4), which was repeated three times, during which they attempted to achieve maximum distance. Then, subjects made a jump with their eyes closed while attempting to reach half of their previous distance. Feedback was then given to the participants regarding the difference between the result in relation to their maximum jump. Following this, the eyes closed jump was repeated a further two times. Results were calculated as “the percentage of error or accuracy in varying the strength according to the formula: the difference between the pattern and the result obtained × 100/50% max result (pattern)” [25] (pp. 152–153).

### 2.4. Statistical Analysis

Statistical analysis was performed using the Statistica 13.1 PL software package (StatSoft Polska, Krakow, Poland). Non-parametric tests were used for analysis due to the size of the groups and skewness of distributions. The Mann–Whitney U-test was employed to analyse differences between Foot-O and T&F groups and between the T&F and Foot-O men’s and women’s groups. The results of the individual tests were ranked from the best to the worst and assigned ranks from 1 to 22. From all four tests, the average rank was calculated and on this basis the results were ranked from the lowest to the highest. Differences between ranks were analysed by the Mann–Whitney U-test. Results were considered statistically significant when *p* ≤ 0.05.

## 3. Results

Table 1 presents the results of the coordination ability tests.

### 3.1. Assessment of Spatial Orientation (T1)

Differences in the average distance (cm) from the goal “x” in the “Walk to the goal” test is presented in column T1. In the Foot-O group, the average distances from the target ranged from 18.8 cm to 59.2 cm, and in the T&F group from 22.2 cm to 69.8 cm. In contrast, women were between 20 cm and 53.6 cm from the target, and men were between 18.8 cm and 59.2 cm. No significant differences were found between the groups in relation to spatial orientation.

### 3.2. Assessment of Rhythmisation (T2)

The subjects’ performance in terms of the difference in time (s) between the 30-metre run test and the run at a given pace across the hoop in the “Run at a given pace” test is shown in column T2. In the Foot-O group, time differences between the two tests ranged from 0.64 s to 1.73 s, and in the T&F group from 0.87 s to 1.82 s, while between the women’s groups from 1.02 s to 1.82 s and the men’s groups from 0.64 s to 1.73 s. No significant differences were found between the groups.

### 3.3. Assessment of Balance (T3)

The results of the “Walk on the balance beam” test for each group are presented in Figure 5. In the Foot-O group, the walking distance covered was from 13 m to 33 m, while in the T&F group it was from 9 m to 29 m. Women walked from 12 m to 29 m, while men walked from 9 m to 33 m. There was a significant difference between the distance covered on the balance beam by the Foot-O group compared to the T&F group (*p* = 0.009) (Table 1, column T3). The Foot-O group covered a much longer distance on the balance beam compared to the T&F group runners.

### 3.4. Assessment of Kinaesthetic Differentiation (T4)

The results of the “Standing long jump from a place at 50% of maximum effort” test are shown in column T4. In the Foot-O group, the percentage of jump accuracy at 50% strength ranged from 67.49% to 97.11%, and in the T&F group from 73.58% to 98.01%. Women achieved a sampling accuracy percentage of 78.87% to 97.99%, and men achieved a sampling accuracy percentage of 67.49% to 98.01%. No significant differences were found between the groups, while a significant difference was found between the T&F and Foot-O men’s groups (*p* = 0.045).

In addition, the results obtained were compared in terms of the average rank of all coordination tests of the Foot-O and T&F groups (Table 2). The study groups differed significantly in the level of motor skills (*p* = 0.011). A significantly better level of coordination motor skills was demonstrated by the Foot-O group.

## 4. Discussion

This study attempted to determine the differences between athletic runners and orienteers in terms of various components of motor coordination. The athletes were subjected to four different tests, each of which assessed slightly different aspects of coordination. Thus, the tests included attempts to determine spatial orientation ability (walk to the goal), rhythmisation (run at a given pace), balance (walk on the balance beam) and kinaesthetic differentiation (standing long jump from a place at 50% of maximum effort).

Spatial orientation was assessed by a trial of marching to a target excluding the sense of sight over a distance of 5 m. All subjects scored between very good and good according to the scale published by Raczek et al. for 13 year olds. The results are slightly worse than those assumed by Thomson [26], who observed that errors during walking distances of 3 m, 6 m and 9 m were up to 24 cm. However, the average trial results are comparable to tests performed over a distance of about 10 m analysed by other authors [27]. No significant gender differences were shown, which is consistent with the researchers’ assertion that there were no gender differences in terms of distance-to-target ratings [28,29,30], although contradictory to the results of many other researchers, the results of which were presented by Coluccia and Louse [31]. The results obtained are in line with those obtained by Pollatou et al. [29], who, after studying 5–6-year-old children, found that there was no relationship between gender and moving “straight ahead” during trials of walking to a target excluding the sense of sight over a short distance (up to 6 m). The results obtained are better in terms of minimum distance from the target “x” than those obtained in the study of Loomis et al. [32] and identical in terms of maximum distance from the target. In the coordination test of walking to target “x” with the exclusion of the sense of sight, it is noted that the results obtained are very similar to those of other researchers [29,32,33,34]. Machowska et al. [35] also compared the differences in the spatial orientation of orienteers vs. athletic runners, but they used blind walking straight ahead on the longer distance (in the airport). These researchers also found no differences between two comparable groups. All of the subjects made smaller or bigger loops instead of going straight. The lack of difference in spatial orientation between athletes and orienteers was probably due to the insufficiently specific test, as orienteers use a completely different type of spatial orientation than the one contained in the battery of tests developed by Raczek et al. [25]. The reason is probably related to turning off the sense of sight, which does not occur in tasks related to the use of spatial orientation by orienteers. Therefore, it would be necessary to develop a specialized test that would test spatial orientation using the sense of sight.

The assessment of rhythmisation did not show any statistical differences in the results between T&F runners and Foot-O. Training in both sports includes many exercises to improve this coordination ability. Competitors make a lot of drills where they put the accent on rhythm. That is why similar training probably leads to these similar effects.

Dynamic balance was measured using a balance beam. The results of our own research indicate significant differences in terms of this test. Those in the Foot-O group travelled a longer distance (m) on the balance beam compared to T&F practitioners, indicating a better performance, which may be due to the orienteers’ frequent movement over varied terrain, or frequent overcoming of terrain obstacles. The distance covered in walking on the balance beam by orienteers is much longer than that observed by Pośpiech et al. [36]. The results show the effect of practicing foot orienteering on the development of balance abilities, which would confirm the results obtained by Jensen et al. [37], Hébert-Losier et al. [38] and Türkmen and Biçer [39]. The latter researchers noted a very clear improvement in both dynamic and static balance after 8 weeks of specialized orienteering training applied to adolescents. Beneficial effects of the influence on the coordination of specialized orienteering training have also been noticed by other scientists [40,41]. Vincent et al. considered orienteering and other trail running in the context of using specialized coordination training to prevent injuries.

The differentiation was measured in the test of mapping 50% of the strength in the jump from a place to a distance. The T&F men’s group performed significantly better compared to the men practicing orienteering. However, the authors were not able to find confirmation in other research results, while the hypothesis is that people who practice athletics are more likely to experience jumping from the spot (performed, for example, during sandbox training). In their training, there are certainly also more exercises that are manifestations of power, such as just jumping from a place, or jumping up.

Ranking the places occupied by Foot-O and T&F in individual trials made it possible to create a ranking of the respondents from the best to the worst results in each test, which in turn made it possible to assess the significance of differences between the results obtained by Foot-O and T&F (the methodology is described in detail in Section 2.4.). Orienteers achieved significantly better results, which indicates better motor coordination as assessed in these four selected tests. The alleged and probable reasons for this advantage result from the factors described below.

Visual information can play an important role in improving motor coordination and its level. It is important for steering and the control of locomotion, particularly over uneven ground and/or avoiding obstacles during everyday life locomotion as well as locomotion in competitive sport contexts such as orienteering and soccer [42]. Perhaps the constant need to control the visual ground and other factors disrupting the course of competition improves motor coordination.

Other researchers have paid attention to yet another factor that has a positive effect on motor coordination [43,44,45]. Mental rotation exercises play an important role in shaping coordination. It turns out that in sports that require mental rotation to perform tasks related to competition (this is the case with orienteering or gymnastics), motor coordination improves. Hurt writes that “arguably, being an orienteer or gymnast involves or requires greater coordination and mental rotation capabilities than being a runner” [44] (p. 14).

Finally, some authors have focused their attention on determining the importance of motor coordination for running efficiency and economy [8,37,46,47]. However, there are different forms of running, while its biomechanics varies depending on the distance required to run, the pace of the run, the slope of the surface on which one runs and the ground on which one runs [37,38,48,49]. This is very easy to observe in orienteers, who change and adapt their running technique to the current conditions. Their running technique looks different when running on hard pavement, and differs when it comes to moving on uneven terrain (such as stone, marsh, etc.). Perhaps this constant need to adapt has developed a better economy of running, which Jensen et al. [37] noted in their study. The conclusion of their study is that running economy was less pronounced in the orienteers than in the track runners, and it could be speculated that specific training may improve running economy, indicating the importance of specific training for orienteers.

## 5. Conclusions

The Foot-O group showed a better level of motor coordination skills, especially balance. Taking into account the analysis broken down by gender, it turned out that men from the T&F group presented a significantly higher level of kinaesthetic differentiation ability.

### Practical Applications and Future Research

Orienteering and orienteers have been the subjects of scientific research many times. The performance of horizontal and vertical running [10,23,50] and types of injuries typical for orienteers [51,52] were assessed as the examples. However, the coordination abilities of orienteers have not been so thoroughly researched. Several researchers [23,39,40] found that all orienteers made significant progress in the development of static and dynamic balance during training, which indicates the influence of specialized orienteering training on the ability to maintain balance. As a result of our research, it can be clearly stated that the ability to maintain balance is undoubtedly a hallmark of orienteers, and perhaps its high level is also necessary for success in this sport. Such a conclusion allows for the preparation of specialized tests for the enrolment or quality control of the training process in the orienteering run.

## Figures and Tables

**Figure 1 ijerph-20-02643-f001:**
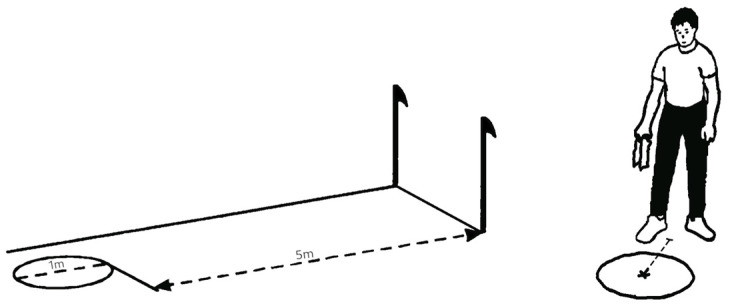
“Walk to the goal” test procedure Raczek et al. 2002 [25] (p. 157). Reprinted with permission. 2023, Wydawnictwo Akademii Wychowania Fizycznego im. J. Kukuczki w Katowicach (Poland).

**Figure 2 ijerph-20-02643-f002:**
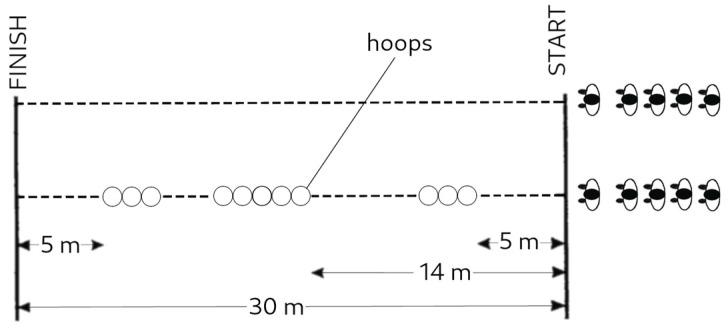
“Run at a given pace” test procedure Raczek et al. 2002 [25] (p. 166). Reprinted with permission. 2023, Wydawnictwo Akademii Wychowania Fizycznego im. J. Kukuczki w Katowicach (Poland).

**Figure 3 ijerph-20-02643-f003:**
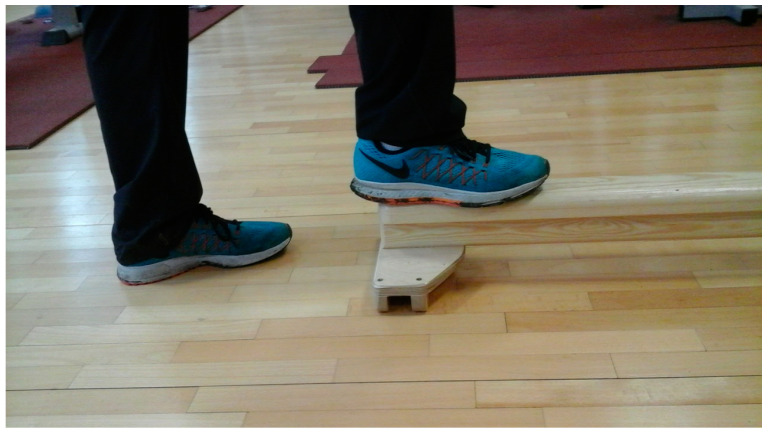
“Walk on the balance beam” start of the test (photo by Weronika Machowska-Krupa).

**Figure 4 ijerph-20-02643-f004:**
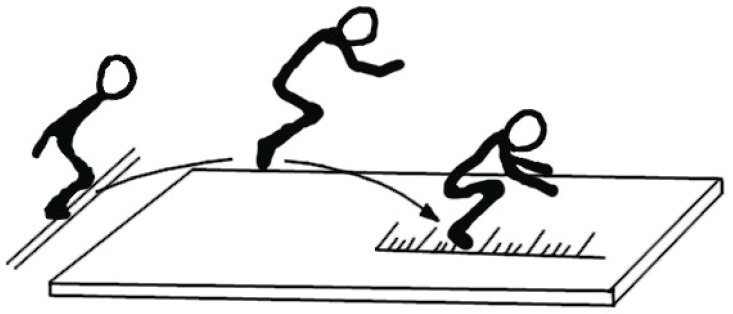
“Standing long jump from a place at 50% of maximum effort” test.

**Figure 5 ijerph-20-02643-f005:**
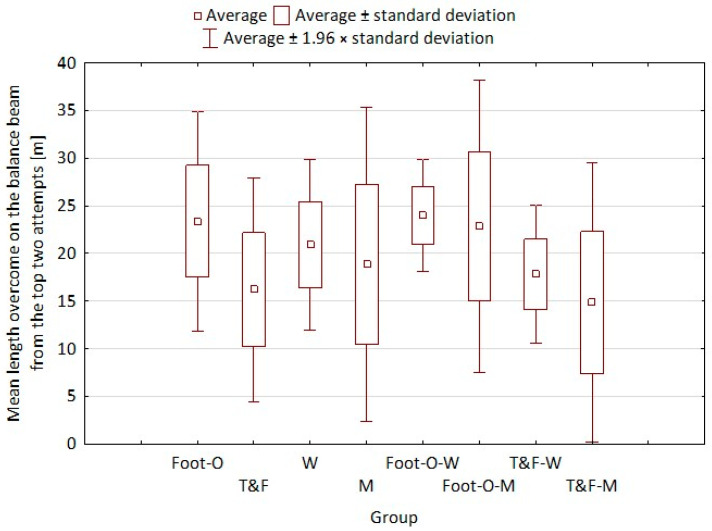
Results from the “Walk on the balance beam” test for each group. Foot-O, f orienteering group; T&F, track and field group; W, women; M, Men; Foot-O-W, women from foot orienteering group; Foot-O-M, men from f orienteering group; T&F-W, women from foot orienteering group; T&F-M, men from foot orienteering group.

**Table 1 ijerph-20-02643-t001:** Results of coordination tests.

Tests																
	T1				T2				T3				T4			
Differences between groups	Mean (SD) (cm)	U	Z	*p*	Mean (SD) (s)	U	Z	*p*	Mean (SD) (m)	U	Z	*p*	Mean (SD) (%)	U	Z	*p*
Foot-O	33.87 (±13.28)	34.50	−1.67	0.094 †	1.32 (±0.30)	54.00	0.39	0.694 †	23.36 (±5.87)	20.50	2.59	0.009 **	84.43 (±9.18)	38.00	−1.45	0.149 †
T&F	45.27 (±13.92)				1.30 (±0.27)				16.18 (±5.98)				89.48 (±7.59)			
Sex differences	Mean (SD) (cm)	U	Z	*p*	Mean (SD) (s)	U	Z	*p*	Mean (SD) (m)	U	Z	*p*	Mean (SD) (%)	U	Z	*p*
Women	39.32 (±10.40)	55.50	0.26	0.792 †	1.38 (±0.26)	50.00	0.63	0.531 †	20.90 (±4.56)	46.50 0.86		0.389 †	88.38 (±6.17)	57.00	0.17	0.869 †
Men	39.78 (±17.69)				1.26 (±0.30)				18.83 (±8.40)				85.77 (±10.36)			

T1, “Walk to the goal” test; T2, “Run at a given pace” test; T3, “Walk on the balance beam” test; T4, “Standing long jump from place at 50% of maximum effort” test; Foot-O, foot orienteering group; T&F, track and field group; U, the Mann–Whitney U test result; Z, the Z test result; *p*, statistical significance; SD, standard deviation; ** significant at the 0.01 probability level; † nonsignificant.

**Table 2 ijerph-20-02643-t002:** Differences between groups in coordination tests (rank).

Differences between Sports	Mean (SD) (Rank)	U	Z	*p*
Foot-O	4.27 (±2.87)	21.50	−2.53	0.011 *
T&F	7.82 (±2.76)			
**Differences between Sexes**	**Mean (SD) (Rank)**	**U**	**Z**	** *p* **
Women	5.50 (±2.98)	51.00	−0.56	0.574 †
Men	6.50 (±3.59)			

Foot-O, foot orienteering group; T&F, track and field group; U, the Mann–Whitney U test result; Z, the Z test result; *p*, statistical significance; SD, standard deviation; * Significant at the 0.05 probability level; † nonsignificant.

## Data Availability

Not applicable.

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
