# Peer review of "Differences in Coordination Motor Abilities between Orienteers and Athletics Runners"

_ijerph, 2023, doi:10.3390/ijerph20032643_

Round 1

Reviewer 1 Report

One concern is the outdated benchmark, no benchmarks from the last 2 years (most current 2 of 2019), need updating.

I am satisfied with the methodology, it describes how the study was carried out, however I recommend reviewing whether it is an experimental study. Where is the intervention done? Is the interview? In my opinion the interview is a method/instrument and not an intervention. Review this.

tables:
organization needs to be improved, mean and standard deviation can be in the same column with XX(XX) and test results should be in column not row; the line is for groups. The same for table 2, there is more space to put the results in columns.

Figure 2: group values are presented and by sex, I suggest putting them in different colors, because there seem to be 4 groups. the figure is fuzzy, the captions and titles are larger than the graph.

Results: the results focus on the test and not on the variable, it is easier to understand when the text refers to the variables and not to the tests. you have to go back and forth to understand the comparisons and differences.

The results are not in line with the objective, was it an objective to compare men and women? needs to be described.

Where is it in the results "level of selected coordination abilities" that is in the objective?

Discussion:
is it necessary to discuss the results, where does the comparison by sex appear? review the need to be featured in the results.
Long paragraphs and again the tests appear, review by variables.

Brief conclusion.

Author Response

Dear Reviewer,
please see the attachment.

Yours faithfully,

authors

Reviewer 2 Report

Dear authors, First of all, congratulate you on your work. Below I will present some comments by way of reflection and suggestion in relation to your study. 1- Table 1 is cut, it is not well understood, in addition to breaking the layout and presentation structure. 2- I would have liked to see the internal differences between Men and Women in the sports modalities themselves I am sure that you have taken this second observation into account and will respond to it in a future work, however, I will comment on it. It only remains to encourage them to continue in this line of research with great zeal.

Author Response

(The authors gave the same response as above.)

Reviewer 3 Report

Differences in Coordination Motor Abilities Between Orienteers and Athletics Runners

This paper reports the results of differences in coordination motor abilities between track and field (T&F) runners and foot-orienteers (Foot O)

Introduction

The introduction is one-sided. It highlighted orienteering but there was no similar information given about track and field. What motor coordination abilities are related to T&F, etc?

Orienteering was said to be beneficial for physical, mental and intellectual prowess. However, no explanation is given how and why.

The authors should cite and introduce literature that relates coordination abilities with orienteering and track and field. The current introduction only discusses coordination in general.

Materials and Methods

Headings are not accurate, e.g. 2.1 Material should be 2.1 Participants, 2.2 Methods should be divided into Measures, Procedures headings. I would also suggest subheadings for each test.

What are the reliability and validity values?

Subjects were blind folded to exclude the sense of sight, but how does that affect balance?

The procedures are not clearly stated, what order was the tests presented? Were they counterbalanced? Any rest intervals? Rest between trials? Were testers blinded on the conditions?

I would like to suggest having diagrams for all tests.

Results

Gender differences were presented in the results but it was not mentioned as a purpose of the study.

For Table 2 and the related information, there is no explanation about the calculation/scoring/formula that combined all the individual test scores to provide the average rank of all coordination trials.

Line 156 how was the analysis for differences between male T&F and male orienteers done? Not specified in 2.3 Statistical analysis

Discussion

Line 167, the study is about track and field, yet the athletic runners are described as treadmill or street. These terminologies must be explained

Line 169, tests or trials?

Line 175, what is the meaning of “running is unequal to running”

Line 187 How does travelling a longer distance on the balance beam indicate better performance which was explained by “the terrain and obstacles”, but in fact the balance beam is flat?

Line 196-97 subjects of which group scored very good and good?

 Discussion on spatial orientation mostly specify which studies are similar or dissimilar without an actual discussion of WHY there were no differences between the orienteers and T&F groups.

There should be a discussion for the average rank of all coordination trials (Table 2).

There should be a discussion for Rhythmisation even though there were no significant differences between groups

Author Response

(The authors gave the same response as above.)

Round 2

Reviewer 3 Report

Thank you for addressing my concerns. The paper is much improved, but I have a few other comments:

The abstract can be expanded to include gender differences as another purpose of the study. Response 9 indicates so, but it has not been added

Abstract can also be expanded to include some information about methods such as demographic information of the participants, number of tests etc.

More importantly, there appears to be errors, e.g. what is the meaning of “interview” in the abstract?

The conclusion can be improved, as the current sentence, “The Foot-O group showed a better level of coordinative motor abilities, with men in the T&F group presenting a significantly higher 326 level of kinaesthetic differentiation abilities”, seems contradictory to say Orienteers is better, but the following explanation is about T&F.

There are some language issues that can be improved, for example, “They also didn’t find”, “Significantly better results scored orienteers”, “described later in the discussion”. Perhaps proofread the whole article again.

Author Response

Dear Reviewer,
We tried to change everything you asked to improve our manuscript. Thank you for your time and all valuable suggestions.
If more changes are needed, we will do whatever you ask.

Sincerely,
authors
